# Research on fire scenario analysis and emergency response strategies for L-shaped buildings using FDS

Qin Zhang🄳[1]*, Yuhong Hu[2], Xiaoju Li[3]

**1** Dongguan City University, Wenchang Road, Liaobu Town, Dongguan, P. R. China, **2** School of Mechanical and Automotive Engineering, South China University of Technology, Wushan Road, Tianhe District, Guangzhou, P. R. China, **3** Dongguan City University, Wenchang Road, Liaobu Town, Dongguan, P. R. China

\* zhangqin@dgcu.edu.cn

## Abstract

With the rapid development of information and communication technologies, modern fire rescue models are evolving towards informatization, proactivity, and spatialization. Existing fire simulation models primarily focus on the impact of individual building parameters, lacking a systematic analysis of the multi-factor coupling effects on fire spread. As a result, it is difficult to accurately predict the trend of fire spread and meet the demands of dynamic escape routes for three-dimensional rescue systems. In this study, a typical L-shaped building was used as the research object, and a parameterized fire scenario database was constructed using the FDS (Fire Dynamics Simulator) fire simulation software, combined with experimental data for multi-factor coupling analysis. By simulating various factors such as initial fire locations (stairwell or corridor corner), floor height of ignition (low/mid/high area), and wind direction, this study systematically reveals the dynamic evolution process of smoke diffusion and the spatial distribution characteristics and interaction mechanisms of fire smoke in indoor and outdoor three-dimensional spaces. It also focuses on the time-dependent variations of smoke propagation in indoor escape routes, staircases, and external walls. The results indicate that the L-shaped building causes asymmetric smoke diffusion. The fire spread mode is a result of multi-factor coupling effects at the fire scene. Safe evacuation decisions are directly related to fire spread characteristics, smoke diffusion paths, the location of trapped individuals, fire alarm occurrence time, and rescue methods. Minor changes in each parameter can lead to significant differences in evacuation outcomes. Based on the situation at the site, generating real-time, dynamic escape paths is a crucial guarantee for improving rescue efficiency and success rates. The findings provide scientific support for the design and optimization of human-machine collaborative rescue systems and offer a data foundation for fire risk assessment and emergency response planning.

**Data availability statement:** All relevant data are within the manuscript and its Supporting Information files.

**Funding:** The author(s) received no specific funding for this work.

**Competing interests:** The authors have declared that no competing interests exist.

## Introduction

As global urbanization continues to accelerate, high-rise and super-tall buildings are increasingly concentrated in dense urban clusters. Owing to their complex spatial configurations and high occupant densities, the complexity of fire prevention and control substantially increases. The average annual growth rate of high-rise building fire incidents worldwide reached 135% between 2023 and 2024. Meanwhile, the deep integration of Internet and Internet of Things (IoT) technologies has fundamentally transformed rescue operations. Trapped occupants have transitioned from uncoordinated self-rescue or passive waiting for external assistance to the active use of smart devices for accessing real-time fire information, dynamic evacuation planning, and self-rescue guidance [1]. In parallel, evacuation strategies have expanded beyond traditional facilities such as staircases and fire ladders to multi-dimensional rescue systems incorporating rescue capsules and descent devices.

To support this emerging rescue paradigm of human-machine collaboration and proactive hazard avoidance, there is an urgent need for a spatial fire evolution prediction model that integrates multiple influencing factors. Fire Dynamics Simulator (FDS), developed by the National Institute of Standards and Technology (NIST) and based on large eddy simulation (LES), has been widely applied for simulating diverse fire scenarios and smoke movement while capturing key performance parameters during fire development [2]. Byström et al. evaluated the reliability of FDS simulation results through full-scale fire experiments [3]. Qiu et al. classified 59 smoke movement models into network, area, field, and hybrid categories and systematically reviewed their computational methodologies [4]. Jones simulated smoke propagation in corridors using regional models and proposed predictive approaches for smoke spread [5]. Tang numerically investigated smoke propagation in straight, circular, L-shaped, and T-shaped corridors and analyzed the influence of corridor geometry on fire performance parameters [6]. Moon et al. conducted full-scale fire simulations for seven types of building fires and three types of indoor combustible materials [7]. Zhang et al. demonstrated that escalator-staircase coupled systems yield higher evacuation efficiency in high-density commercial buildings [8]. Yan et al. showed that coordinated utilization of staircases, elevators, and refuge floors significantly improves evacuation efficiency in super high-rise buildings [9], and further analyzed optimal evacuation strategies under different occupancy conditions [10,11].

An analysis of the current state of fire simulation reveals clear limitations in existing fire simulation models. First, the scenarios are overly simplistic, with current simulations focusing on the independent effects of single building parameters (such as corridor structure), lacking a systematic study of the coupling effects of multiple factors like fire point spatial distribution, floor height, and external wind fields. Secondly, there is a lack of dynamic prediction; no real-time prediction model for indoor safety evacuation time (RSET) and safe areas on building exterior walls has been established. As a result, simulation outcomes are difficult to directly apply to dynamic decision-making in active three-dimensional rescue systems. To address these issues, this study proposes a multi-scenario coupled fire evolution simulation framework, taking L-shaped buildings as an example.

L-shaped buildings are common representative asymmetric structures, with their corner structures easily creating special airflow patterns that affect the spread of smoke. This study aims to build a multi-factor coupled fire evolution prediction model to forecast fire trends, providing a theoretical basis for generating spatial and dynamic escape routes of trapped individuals in different locations within L-shaped buildings, thereby improving rescue efficiency for trapped individuals.

## Establishing the fire model

Existing fire simulation studies primarily focus on regular building layouts, lacking scene-based frameworks for complex geometries such as L-shaped buildings. To reveal the fire evolution patterns in L-shaped buildings and quantify the coupling effects of three key factors- initial fire locations (stairwell or corridor corner), floor height of ignition (low/mid/high area), and wind direction-on fire spread inside the building and along exterior walls, this study establishes a fire model for a 15-story L-shaped building. The building is assumed to be a reinforced concrete structure with a fire resistance rating of Class II according to Chinese standards. The building's total front length is 45.5 meters, the side length is 45.5 meters, the floor height is 3.9 meters, and the total building height is 58.5 meters. The floor distribution for the standard floor is shown in Fig 1(a). Each floor can be divided into 24 rooms, 2 stairwell areas, and 3 corridor areas. The elevator shaft is not depicted in the building floor plan. The first floor has three additional safety escape doors, marked in purple in the diagram, including one main entrance and two side doors. The corridor areas are divided into Corridor Area A (pink), Corridor Area B (gray), and Corridor Area C (blue), with a corridor width of 1.8 meters.

## Fire scenario parameter design

Following the worst-case scenario principle, fire source locations were selected at representative positions anticipated to yield the most rapid fire spread and severe hazards. A fire originating on the ground floor, which features multiple exits and serves as a critical evacuation path, can propagate swiftly. A fire on a middle floor may hinder evacuation on both upper and lower levels, while a fire on the top floor is highly susceptible to the stack effect, complicating spatial rescue operations. Consequently, for the 15-story high-rise building under investigation, three representative floors were designated as ignition levels: the ground floor (1st floor), a middle floor (8th floor), and the top floor (15th floor).

The corridor outside the room nearest to staircases or the building corner was identified as a critical segment of the evacuation path. An ignition point near a staircase could compromise its functionality for egress, whereas a point near a corner could be significantly influenced by local ventilation conditions. Accordingly, three representative ignition points were defined: Initial fire location ① in Room 13, closest to Staircase 1; Initial fire location ② in Room 17, nearest to the L-shaped building corner; and Initial fire location ③ in Room 22, closest to Staircase 2. These three points, marked by red squares "■" in Fig 1(a), were utilized to investigate the impact of fire source location on fire spread dynamics.

Wind direction is a critical factor influencing fire spread. External wind was introduced from four orthogonal directions relative to the L-shaped building (front, rear, left, and right), as indicated by the arrows in Fig 1(a). This setup was designed to test smoke dispersion under conditions where the building was windward, leeward, laterally exposed without obstruction, and laterally shielded, thereby examining the effect of wind direction on fire development.

The building for simulation was defined as a sprinkler-equipped office building, with robotic rescue capsules deployed on the rooftop. Based on the typical heat release rate for office fires (GB 51251−2017), the fire source was set to a constant heat release rate of 1.5 MW. To simulate the fire resistance characteristics of reinforced concrete structures meeting the Chinese Grade II fire resistance rating requirements (GB 50016−2014), building surface materials were specified as inert, with a 0.1 m thick gypsum layer. The combustion reaction was modeled on polyurethane, a representative flammable material found in offices.

Given that the multi-year average wind speed at a 10 m height in Guangzhou, China is 1.9 m / s, which reflects the regional wind environment, a velocity ($V_0$) of 1.9 m / s at a reference height ($H_0$) of 10 m was adopted for the simulations.

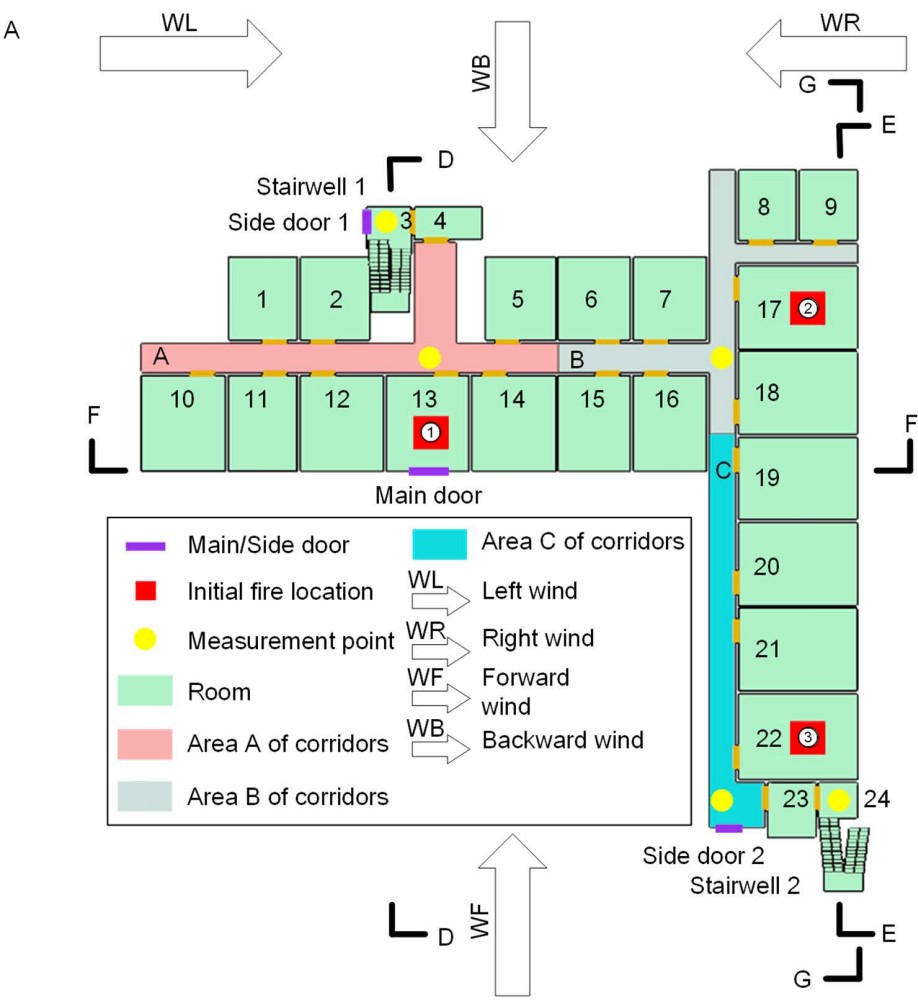

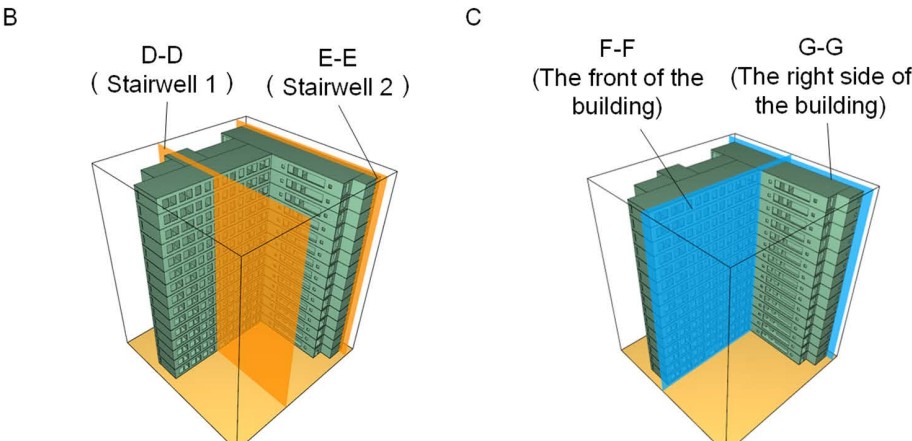

**Fig 1. Fire-simulated model of an L-shaped building. (A)** Floor plan. **(B)** D-D and E-E sections. **(C)** F-F and G-G views.

Simulations were conducted to evaluate the effects of floor height of ignition, initial fire locations, and wind direction on fire spread. The detailed configurations for all scenarios are summarized in Table 1.

To ensure the generalizability of the study, two distinct window states were simulated for each fire scenario case: All Windows Normally Closed: The fire initially spreads indoors. Windows were set to break when the temperature exceeded 250°C, allowing flames and smoke to vent to the building exterior. This scenario represents the fastest indoor fire spread and highest indoor hazard, suitable for simulating internal fire development processes. All Windows Normally Open: The fire spreads rapidly to the external wall immediately upon ignition, representing the greatest impact on the facade and the highest external hazard. This scenario was used to characterize external wall fire spread [11]. Key fire hazard parameters, namely smoke layer height, smoke temperature, visibility, and CO concentration, were selected for analysis. The performance-based assessment criteria adopted for evaluating occupant safe evacuation were summarized in Table 2. [11]

To monitor key parameters along evacuation routes during fire propagation, measurement points were defined in the FDS simulations at locations marked by yellow circles "●" in Fig 1(a). These points were distributed within Corridor Areas A, B, and C, as well as Stairwells 1 and 2, on each storey of the L-shaped building. At each measurement location, smoke layer height and the temperature, visibility, and CO concentration were recorded at 2 m above floor level, thereby characterizing local smoke propagation conditions along evacuation routes. To capture smoke propagation along the inner and outer wall surfaces of the evacuation stairwells across multiple storeys, vertical performance parameter profiles D-D and E-E were defined for Stairwells 1 and 2, respectively, as illustrated in Fig 1(b). In addition, performance parameter sections F-F and G-G were defined at the front and right-hand exterior facades of the L-shaped building, as depicted in Fig 1(c). These profiles describe the vertical progression of fire and smoke along the external walls at different storeys. Taking into account the dual considerations of simulation accuracy and computational efficiency, a uniform grid size of 0.5 m × 0.5 m × 0.5 m was selected as the design value for the computational model and was adopted for all subsequent numerical simulations and experimental investigations.

**Table 1. Different Fire Scenarios of an L-shaped building.**

| Fire Scenario Case | Initial Fire Location | Fire Floor | Wind Direction | Window Status |
|---|---|---|---|---|
| 1 | ① | 8th floor | None | Normally closed, open at 250°C |
| | | 8th floor | None | Normally open |
| 2 | ② | 8th floor | None | Normally closed, open at 250°C |
| | | 1st floor | None | Normally open |
| 3 | ③ | 15th floor | None | Normally closed, open at 250°C |
| | | 8th floor | None | Normally open |
| 4 | ① | 8th floor | None | Normally closed, open at 250°C |
| | | 8th floor | None | Normally open |
| 5 | ① | 8th floor | None | Normally closed, open at 250°C |
| | | 8th floor | | Normally open |
| 6 | ① | 8th floor | WL | Normally closed, open at 250°C |
| | | 8th floor | | Normally open |
| 7 | ① | 1st floor | WR | Normally closed, open at 250°C |
| | | 15th floor | | Normally open |
| 8 | ① | 8th floor | WF | Normally closed, open at 250°C |
| | | 8th floor | | Normally open |
| 9 | ① | 8th floor | WB | Normally closed, open at 250°C |
| | | | | Normally open |

**Table 2. Performance Parameters for Safe Evacuation.**

| Properties | Allowable limit |
|---|---|
| Smoke layer height | $\geq$ 2 m |
| Temperature at 2 m above the ground | $\leq$ 60°C |
| Visibility at 2 m above the ground | $\geq$ 10 m |
| CO concentration at 2 m above the ground | $\geq$ 500ppm |

## Selection of performance parameters

To identify an appropriate fire parameter for safety assessment, Case 1-defined by ignition in Room 13 adjacent to Stairwell 1 on the 8th floor (mid-level)-was selected as the control scenario. Fire spread characteristics within the fire floor, stairwells on each level, and along external walls were analyzed under the various scenarios summarized in Table 1, within a rescue time window of 600 s. Fig 2 illustrates the temporal evolution of fire parameters on the fire floor during the 600 s rescue period, under conditions where all windows were initially closed and subsequently fractured at 250°C due to excessive temperatures. This scenario represents the most rapid indoor fire development and the highest hazard level. Within the simulation duration, visibility at 2 m above floor level in Corridor Areas A, B, and C, as well as Stairwells 1 and 2, rapidly decreased to hazardous levels, as shown in Fig 2(c). In contrast, other fire parameters associated with the smoke layer, including temperature, smoke layer height, and CO concentration (Fig 2(a), (b), and (d)), remained below critical thresholds throughout the same period. Figs 3 and 4 illustrate the temporal evolution of fire parameters in Stairwells 1 and 2 on floors adjacent to the ignition level within the 600 s simulation period. Visibility on the fire floor and the three floors above it (Floors 8–11) in Stairwell 1, as well as on the fire floor and the floor immediately below it (Floors 8 and 7) in Stairwell 2, progressively reached hazardous levels over time. Figs 5 and 6 present fire parameter distributions along sections D-D and E-E of Stairwells 1 and 2, respectively, at 600 s in Case 1. It was observed that under closed-window conditions followed by glass breakage at 250°C, visibility in Stairwells 1 and 2, Corridor Area A, and corresponding areas on adjacent floors reached hazardous levels.

When all windows remained fully open, the fire spread rapidly toward the exterior wall surfaces, exerting the greatest impact on the external structure and presenting the highest level of external fire hazard. Fig 7 presents the smoke propagation pattern along the external wall at 600 s under fully open window conditions in Case 1, indicating that smoke accumulation along the facade was most pronounced in this scenario. Figs 8 and 9 illustrate fire parameter distributions at 600 s along the front elevation (F-F) and right elevation (G-G) of the L-shaped building, respectively. Upward fire and smoke

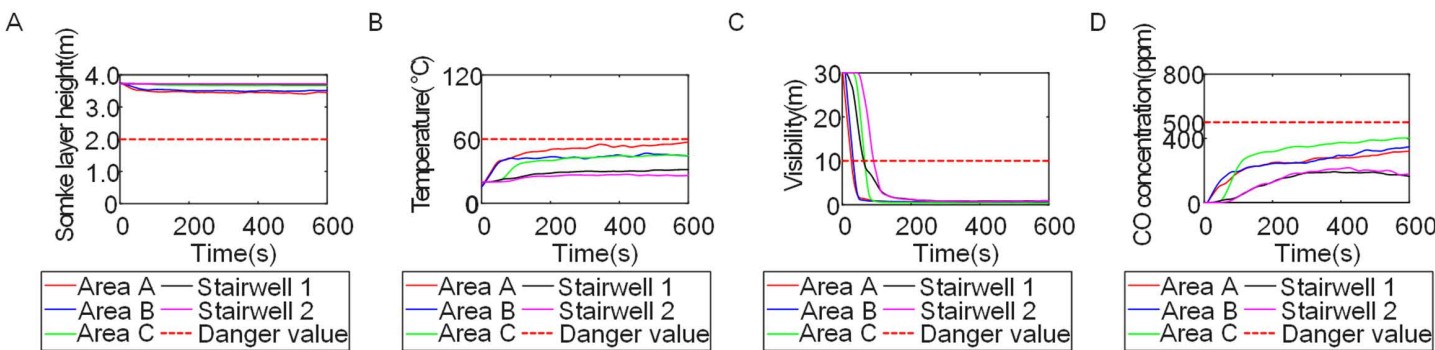

**Fig 2. Performance parameters of the fire floor in Case 1 during the 600s. (A)** Smoke layer height over time. **(B)** Temperature over time. **(C)** Visibility over time. **(D)** CO concentration over time.

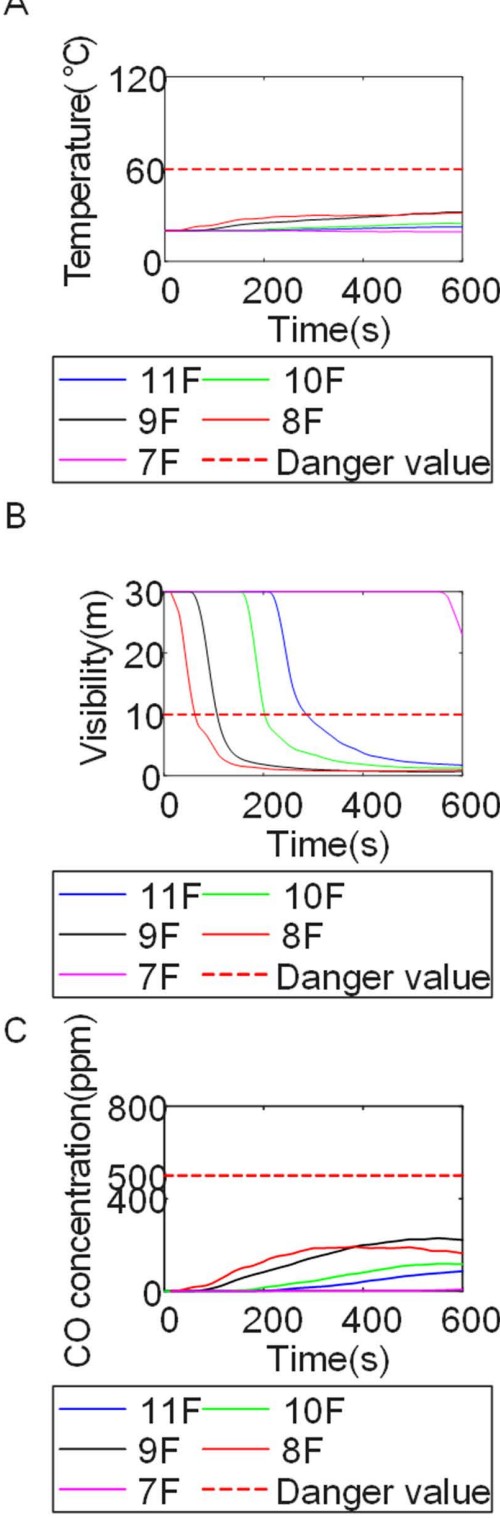

**Fig 3. Performance parameters at the stairwell 1 location in Case 1 over time. (A)** Temperature distribution. **(B)** Visibility distribution. **(C)** CO concentration distribution.

A

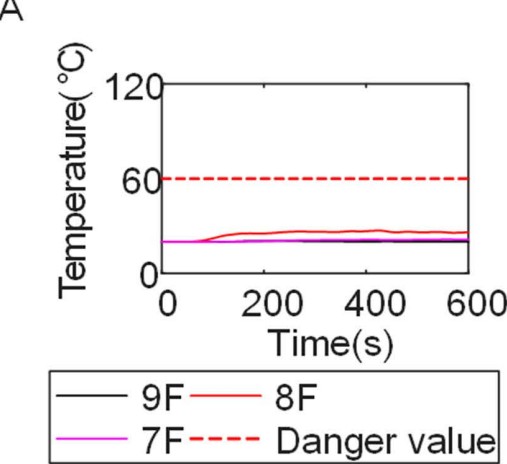

B

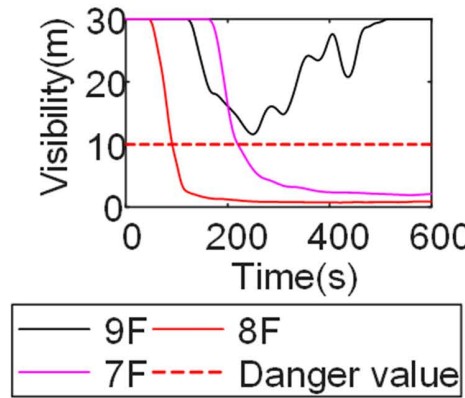

C

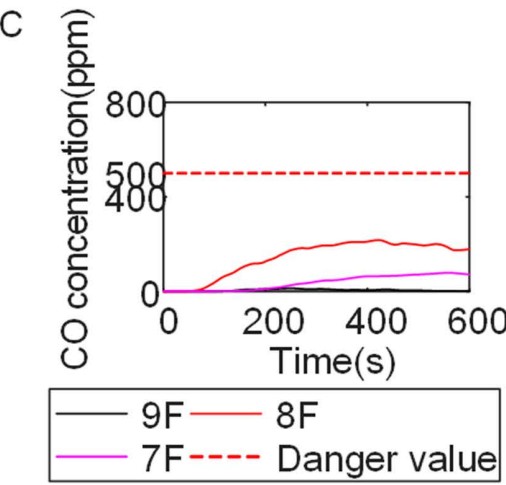

**Fig 4. Performance parameters at the stairwell 2 location in Case 1 over time. (A)** Temperature distribution. **(B)** Visibility distribution.
**(C)** CO concentration distribution.

Stairwell 1

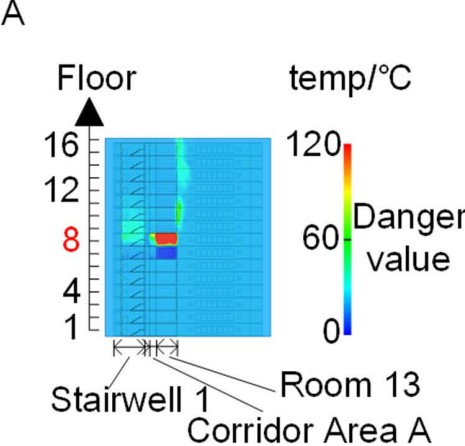

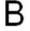

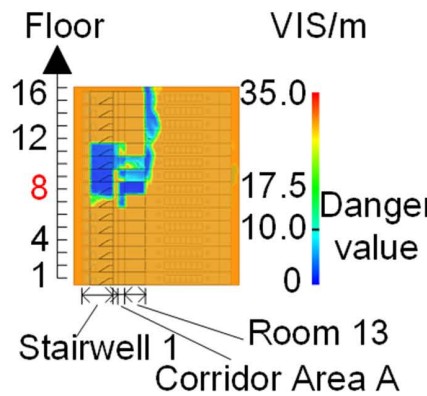

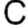

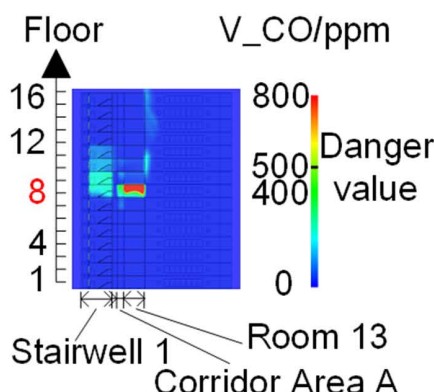

**Fig 5. Fire parameters for the D-D section in Case 1. (A)** Temperature distribution. **(B)** Visibility distribution. **(C)** CO concentration distribution.

Stairwell 2

A

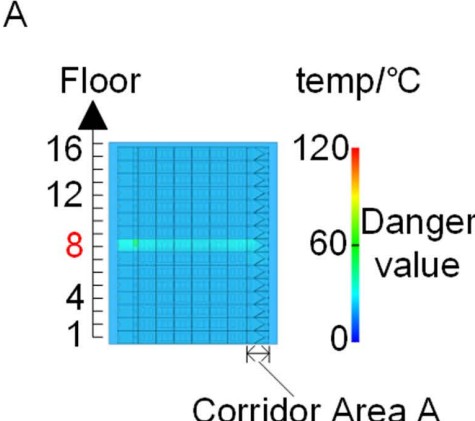

B

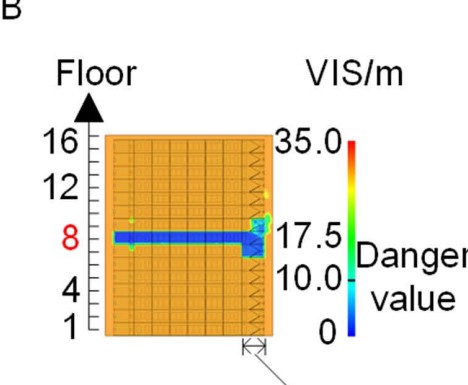

C

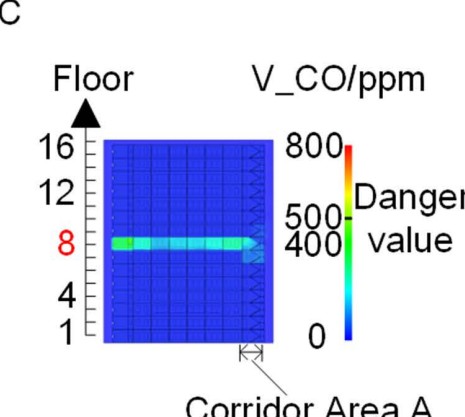

**Fig 6. Fire parameters for the E-E section in Case 1. (A)** Temperature distribution. **(B)** Visibility distribution. **(C)** CO concentration distribution.

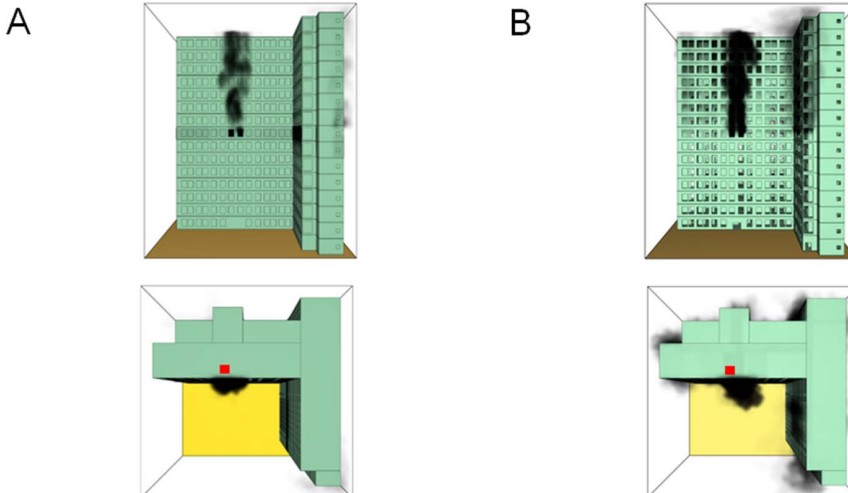

**Fig 7. Live images in Case 1at 600 seconds (red squares indicate initial fire location). (A)** Windows kept closed, opened at 250°C. **(B)** Windows kept open.

spread along the external walls driven by the chimney effect was clearly observed. Compared with temperature and CO concentration, which affected relatively limited regions, visibility exhibited a substantially wider hazardous influence range within the same simulation period.

Based on the above analyses of fire spread within the building-including the fire floor and stairwells-and along the external walls in Case 1, visibility was identified as the most sensitive indicator of fire propagation among the considered parameters (smoke layer height, temperature, and CO concentration). Visibility responded more rapidly to changes in fire development and more effectively reflected on-site fire conditions. Accordingly, in subsequent comparative analyses, the visibility threshold corresponding to the most hazardous indoor fire scenario-where windows fractured at 250°C-was selected as the primary assessment criterion for indoor fire propagation. For external wall fire spread, visibility under the most unfavorable condition, with all windows fully open, was adopted as the benchmark criterion.

## The influence of building parameters on fire spread

### Effect of initial fire location

To investigate the influence of initial fire location on fire spread, Cases 2 and 3 were selected as comparative scenarios, with ignition points located in Room 17 adjacent to the L-shaped corner and Room 22 adjacent to Stairwell 2, respectively. Fig 10 presents the temporal evolution of visibility at different locations on the fire floor as the initial fire location varied. Compared with Case 1 (Fig 2), distinct fire propagation trends were observed following changes in initial fire location. In Case 2, Stairwell 1 on the fire floor remained tenable for the entire 600 s simulation period, whereas Stairwell 2 became untenable after approximately 60 s. In Case 3, Stairwell 1 remained tenable for approximately 300 s, while Stairwell 2 reached hazardous visibility levels within 50 s. By contrast, in Case 1, Stairwell 1 remained tenable for only 60 s and Staircase 2 for approximately 120 s. These results indicate that evacuation routes on the fire floor are highly dependent on the initial fire location. Accordingly, occupants on the fire floor should evacuate via the stairwell that remains tenable within the corresponding time window.

Figs 11 and 12 illustrate visibility evolution in Stairwells 1 and 2 on floors adjacent to the fire floor within the 600 s simulation period. As can be seen from Figs 11 and 12, because the initial fire locations in Cases 2 and 3 were

The front of the building

**A**

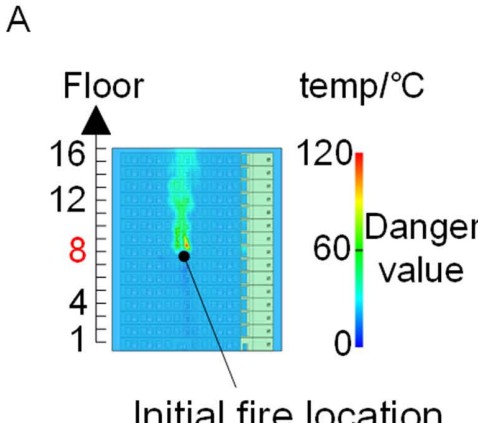

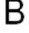

**B**

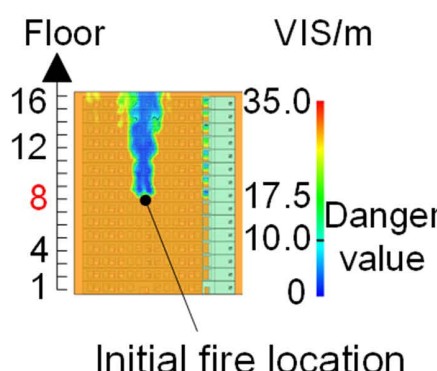

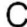

**C**

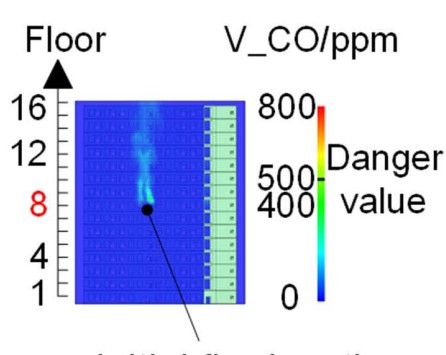

**Fig 8. Fire parameters for the F-F section in Case 1. (A)** Temperature distribution. **(B)** Visibility distribution. **(C)** CO concentration distribution.

The right side of the building

A

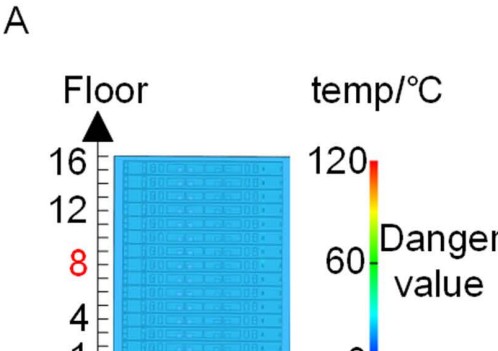

B

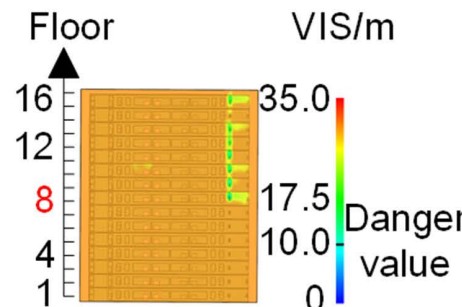

C

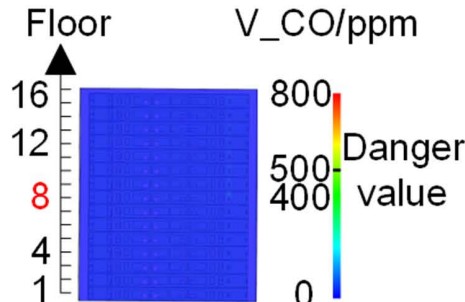

**Fig 9. Fire parameters for the G-G section in Case 1. (A)** Temperature distribution. **(B)** Visibility distribution. **(C)** CO concentration distribution.

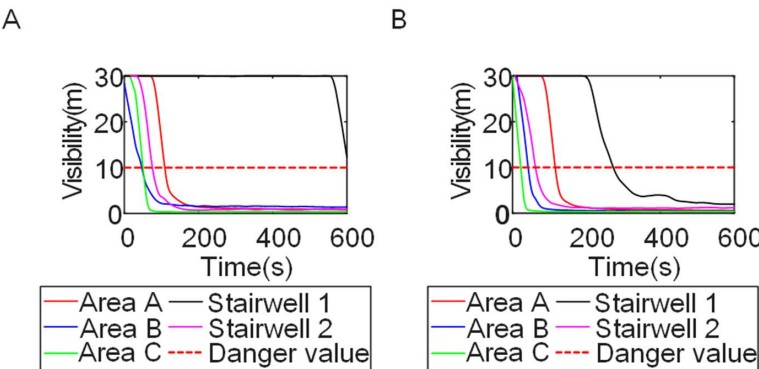

**Fig 10. Visibility on the fire floor over time. (A)** Case2. **(B)** Case3.

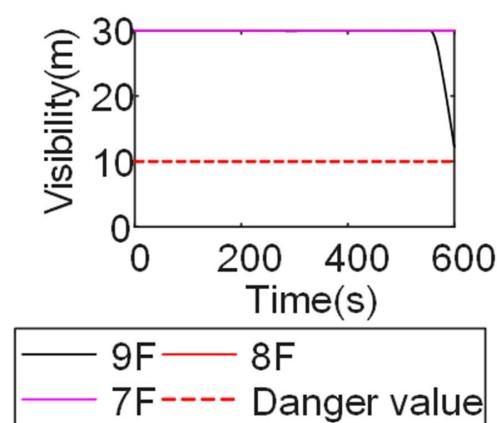

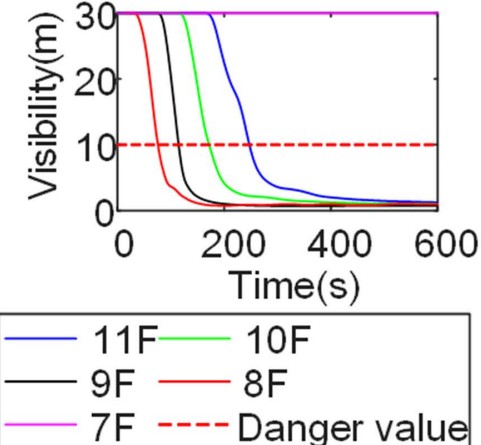

**Fig 11. Visibility in Stairwells in Case 2 over time. (A)** Case2 Stairwell 1. **(B)** Case2 Stairwell 2.

A

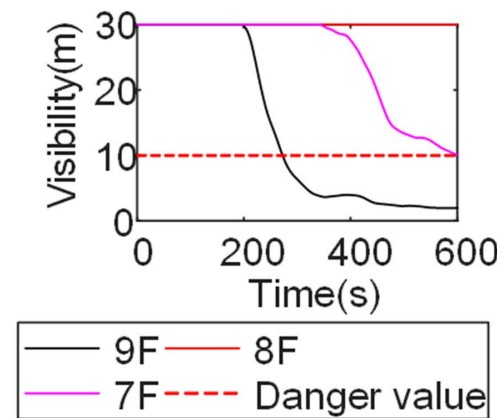

B

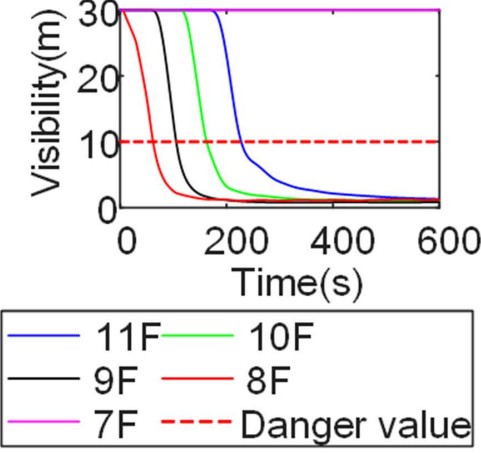

**Fig 12. Visibility in Stairwells in Case 3 over time. (A)** Case3 Stairwell 1. **(B)** Case3 Stairwell 2.

located farther from Stairwell 1, visibility degradation in Stairwell 1 on the fire floor and on adjacent floors above and below occurred more gradually. In contrast, visibility in Stairwell 2 on the fire floor and on the three floors above it rapidly reached hazardous levels, while all floors below the fire floor remained tenable. Under these conditions, occupants on the fire floor could evacuate downward via Staircase 1, which remained readily accessible. Comparing the analysis of visibility changes over time within 600 seconds for each floor's stairwell in Case 1, as shown in Fig 3(b) and Fig 4(b), it further demonstrates that different initial fire locations result in distinct optimal evacuation routes within the building. When the ignition point corresponded to initial fire location ② and ③ (Cases 2 and 3), evacuation via Staircase 1 was consistently the safest option. When the ignition point corresponded to initial fire location ① (Case 1), evacuation had to be completed within approximately 60 s via Staircase 1 to a lower floor, or within approximately 100 s by moving upward to an upper floor followed by external rescue using a rescue capsule. Fig 13 presents instantaneous smoke distribution snapshots at 600 s. In Cases 2 and 3, smoke propagation was primarily

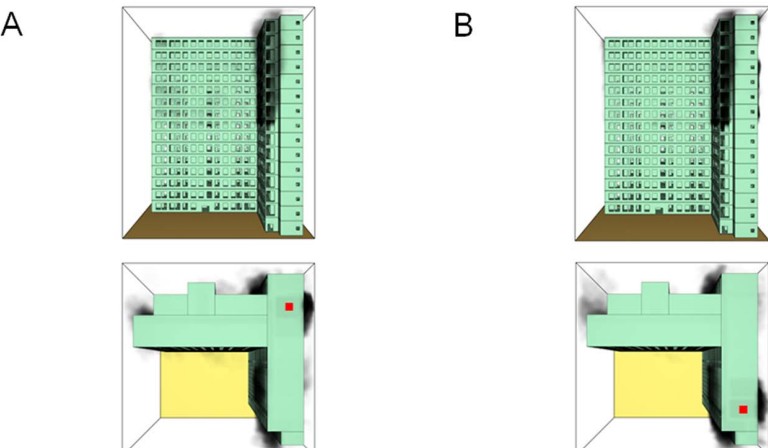

**Fig 13. Live images at 600 seconds (red squares indicate initial fire location). (A)** Case2 Live images. **(B)** Case3 Live images.

concentrated along the right-hand side of the L-shaped building near initial fire locations ② and ③. Smoke preferentially propagated along the straight corridor on the fire floor. Figs 14 and 15 present visibility distributions along sections F-F and G-G near the external walls at 600 s for Cases 2 and 3, respectively. Comparison with the corresponding results for Case 1 (Fig 8(b) and Fig 9(b)) indicates that the spatial extent of hazardous visibility on the front and right-hand facade s of the L-shaped building was strongly correlated with the initial fire location. During fire spread along external walls, the hazardous height range generally extended from the fire floor to the top floor. To enhance evacuation effectiveness, the placement of external collective rescue capsules should be optimized by locating them on building facade s distant from the initial fire location and adjacent to rooms near evacuation stairwells. For example, in Case 2-where Stairwell 1 remained tenable for an extended period-the collective rescue capsule was positioned outside Room 13 on the front facade of the building, adjacent to Stairwell 1.

### Effect of the location of the fire floor

Cases 4, 5, and 1-corresponding to fire floors at the ground (1st), top (15th), and middle (8th) levels, respectively-were selected for comparison to examine fire propagation across the fire floor, within stairwells on each level, and along external walls, and to evaluate how the fire floor location influences key fire parameters. Comparison of the visibility curves in Fig 16 for Cases 4 and 5 with those in Fig 2(c) for Case 1 shows distinct fire spread characteristics associated with different fire floor locations. In Case 4, where the fire originated on the ground floor, the initial fire location was adjacent to the normally open Main door, while Stairwells 1 and 2 were adjacent to Side doors 1 and 2, respectively. Consequently, smoke propagation along the corridor was accompanied by outward discharge through the Main door. As a result, despite its proximity to the initial fire location, Stairwell 1 experienced the slowest visibility degradation. The onset of visibility degradation in Corridor Area C and Stairwell 2-located farther from the initial fire location-occurred significantly later than that observed in the corresponding areas in Case 1. In Case 5, with the fire located on the top floor, upward smoke migration through the stairwells was restricted. Smoke therefore accumulated primarily within rooms adjacent to Corridor Areas A, B, and C, resulting in limited vertical smoke spread beyond the fire floor.

Figs 17 and 18 illustrate visibility evolution in Stairwells 1 and 2 on floors adjacent to the fire floor within the 600 s simulation period. Compared with Case 1 (Fig 3(b) and Fig 4(b)), where the fire originated on the middle floor, smoke propagation within Stairwells 1 and 2 occurred more rapidly on the fire floor. In Stairwell 1, located

A

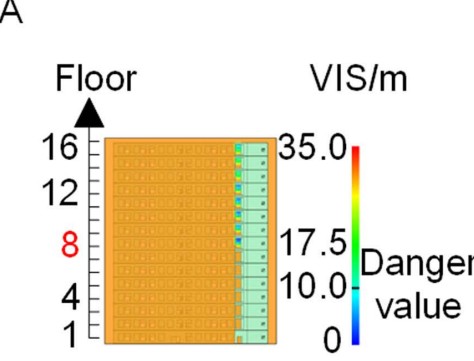

B

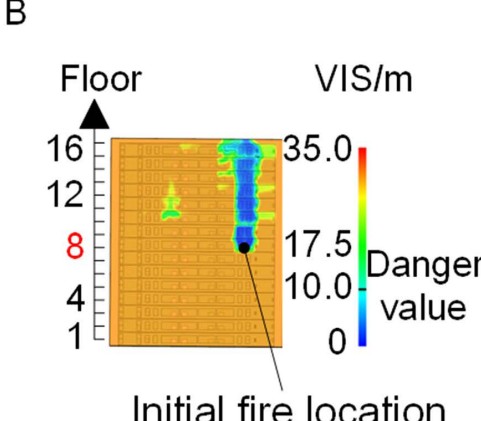

Initial fire location

**Fig 14. Visibility of building exteriors in Case 2. (A)** Case2 F-F section. **(B)** Case2 G-G section.

near the initial fire location, smoke predominantly propagated upward along the stairwell, significantly affecting the fire floor and upper floors. In contrast, smoke in Stairwell 2, which was farther from the initial fire location, propagated both upward and downward along the fire floor. Fig 19 presents instantaneous smoke distribution snapshots at 600 s. Figs 20 and 21 illustrate fire spread along the external walls for different fire floor locations at the same time point. In Case 4, smoke accumulation was concentrated near the Main door, Side door 1, and Side door 2-that is, along the front, left, and rear facade s of the building-and extended vertically from the ground floor to the top floor. In Case 5, smoke spread was primarily concentrated near the front facade adjacent to the initial fire location and remained confined to the vicinity of the top floor. Comparison with Case 1 (Fig 8(b) and Fig 9(b)) indicates that the spatial extent of hazardous visibility on the front and right-hand facade s was strongly dependent on the fire floor location. Moreover, a funnel-shaped vertical diffusion pattern was observed along the external walls. To enhance evacuation effectiveness, the placement of collective rescue capsules should be optimized by positioning them, as far as practicable, within the rescue area below the fire floor, where visibility conditions remain tenable.

A

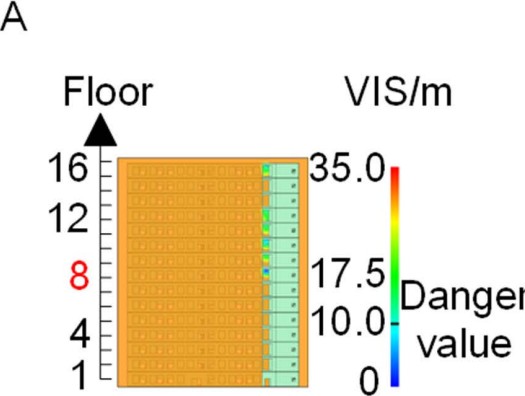

B

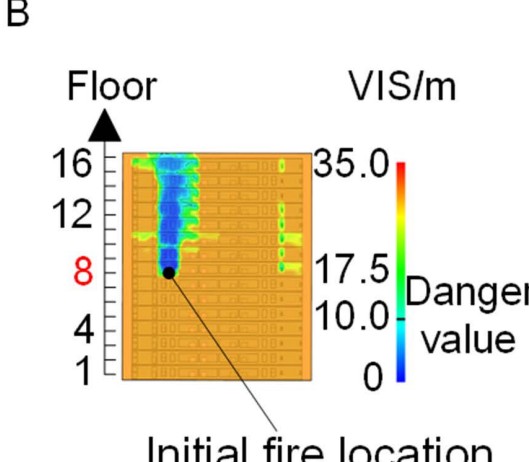

Initial fire location

**Fig 15. Visibility of building exteriors in Case 3. (A)** Case3 F-F section. **(B)** Case3 G-G section.

### Effect of wind direction

The influence of wind direction on fire spread was investigated using fire scenarios with varying wind conditions (Cases 6–9). Fig 22 shows that changes in wind direction resulted in distinct visibility distributions across different regions of the fire floor. Variations in wind direction altered the pressure field governing smoke dispersion, thereby modifying smoke propagation pathways within the building. The temporal evolution of visibility in stairwells adjacent to the fire floor is presented in Figs 23–26. These results demonstrate that wind direction strongly influences the direction and pattern of fire and smoke spread. Fig 27 presents instantaneous smoke distribution snapshots at 600 s under different wind directions. Figs 28–31 illustrate the spread of fire along the external walls at 600 seconds under different wind directions. In Case 6, upward airflow induced by obstruction from the right-hand building altered smoke dispersion patterns within the L-shaped structure. Under leftward wind pressure, smoke initially dispersed toward the left near the fire floor and subsequently shifted rightward at higher elevations. As indicated in Fig 23, occupants on the fire floor were able to evacuate downward

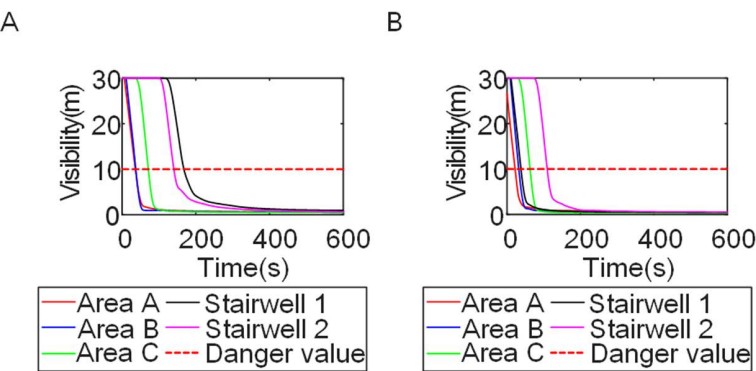

**Fig 16. Visibility on the fire floor over time. (A)** Case4. **(B)** Case5.

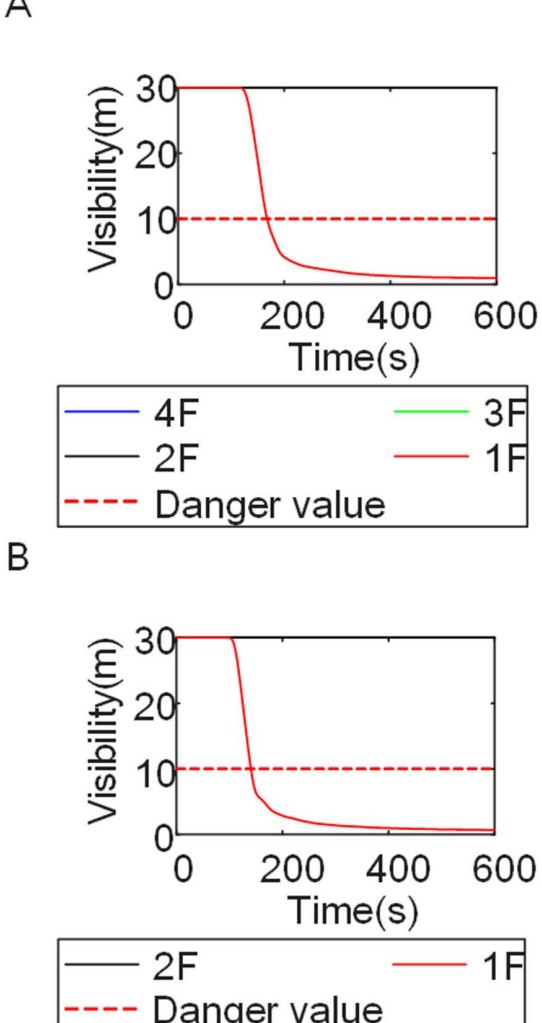

**Fig 17. Visibility in Stairwells in Case 4 over time. (A)** Case4 Stairwell 1. **(B)** Case4 Stairwell 2.

A

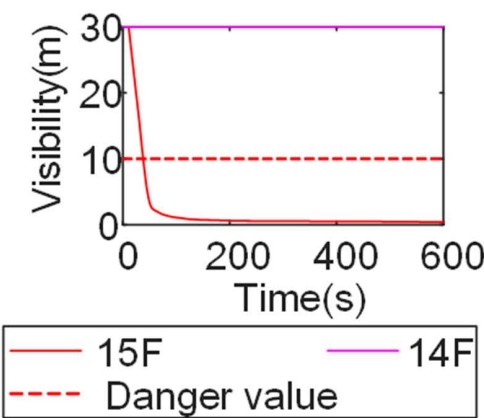

B

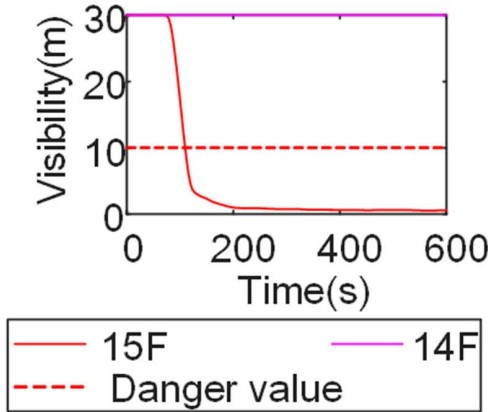

**Fig 18. Visibility in Stairwells in Case 5 over time. (A)** Case5 Stairwell 1. **(B)** Case5 Stairwell 2.

within 300 s via either Stairwell 1 or Stairwell 2. Occupants located above the fire floor could evacuate via rescue capsules accessed through Stairwell 2.

In Case 7, smoke initially propagated toward the right near the fire floor under rightward wind pressure and subsequently dispersed toward the left at higher elevations. As shown in Fig 24, occupants on floors adjacent to the fire floor could evacuate downward via Stairwell 2 or ascend to upper floors via Stairwell 1 to access rescue capsules for evacuation. In Cases 8 and 9, smoke propagation predominantly followed the prevailing wind direction when subjected to frontal and backward wind forces. Based on the above results, comparison with the visibility slice distributions for Case 1 (Fig 8(b)and Fig 9(b)) indicates that the spatial extent of hazardous visibility on the front and right-hand facade s of the L-shaped building was strongly dependent on wind direction. Under frontal and backward wind conditions, smoke dispersion primarily followed the wind direction. When the initial fire point is located on the windward side, smoke dispersion initially occurred against the wind direction due to pressure effects and subsequently aligned with the wind at higher elevations. Under windy conditions, collective rescue capsules should be preferentially

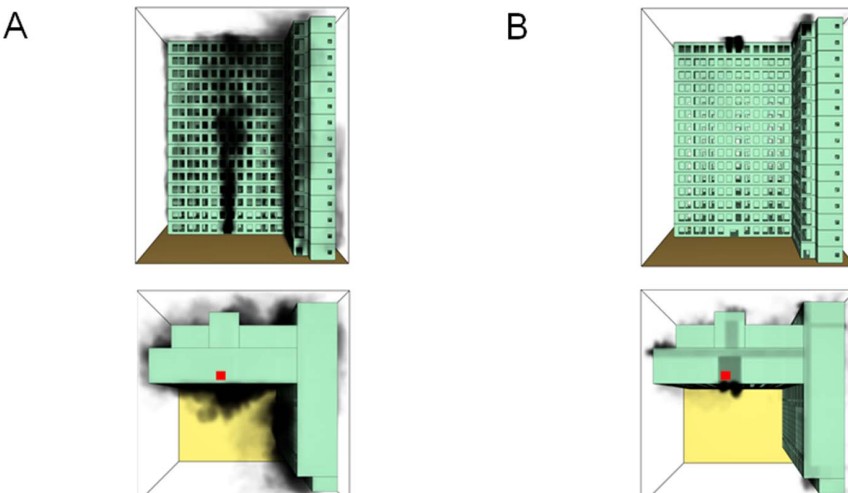

**Fig 19. Live images at 600 seconds (red squares indicate initial fire location). (A)** Case4 Live images. **(B)** Case5 Live images.

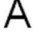https://doi.org/10.1371/journal.pone.0346927.g019

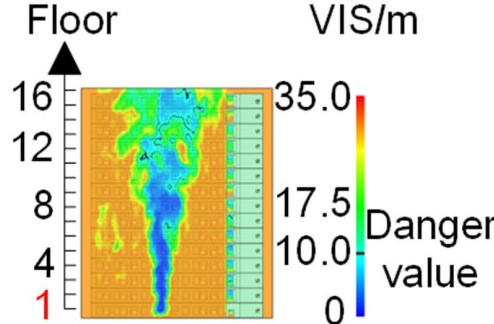

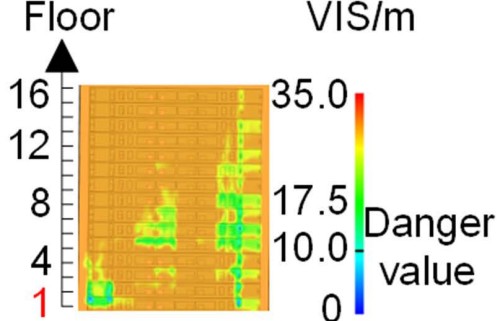

**Fig 20. Visibility of building exteriors in Case 4. (A)** Case 4 F-F section. **(B)** Case 4 G-G section.

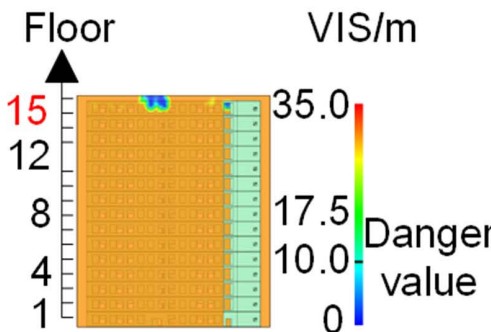

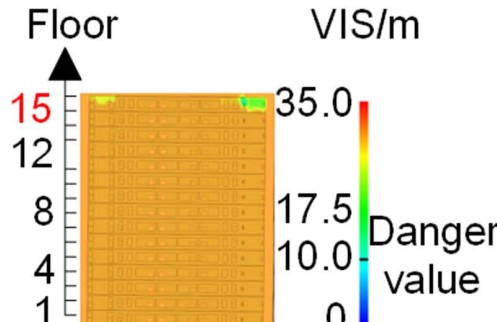

**Fig 21. Visibility of building exteriors in Case 5. (A)** Case 5 F-F section. **(B)** Case 5 G-G section.

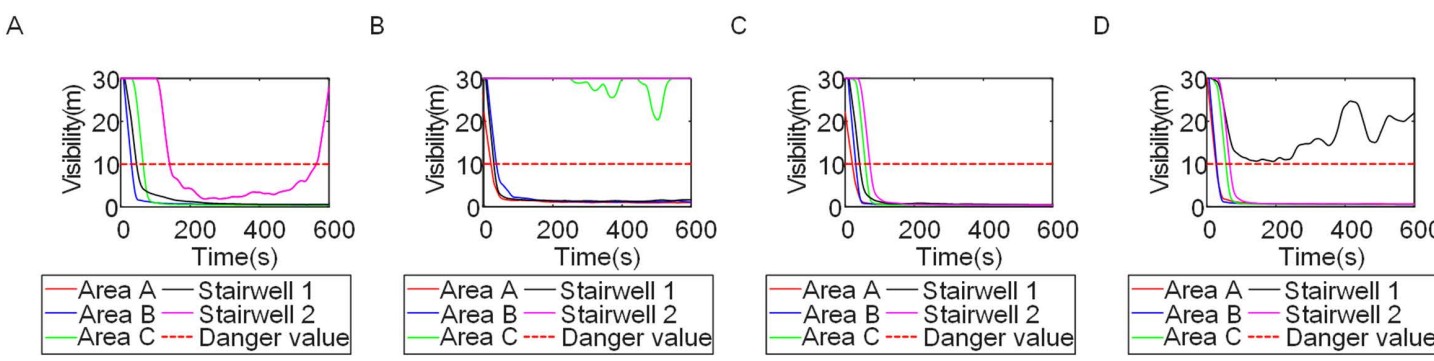

**Fig 22. Visibility on the fire floor over time. (A)** Case6. **(B)** Case7. **(C)** Case8. **(D)** Case9.

A

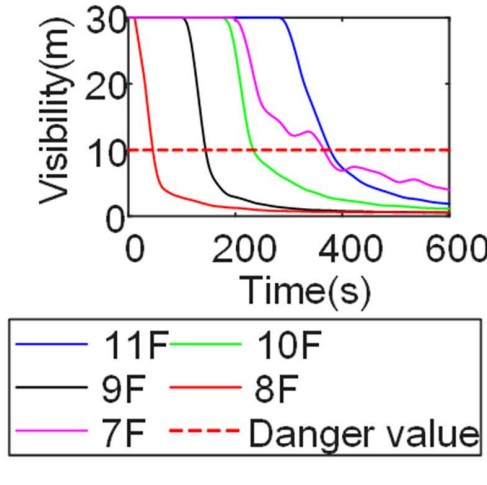

B

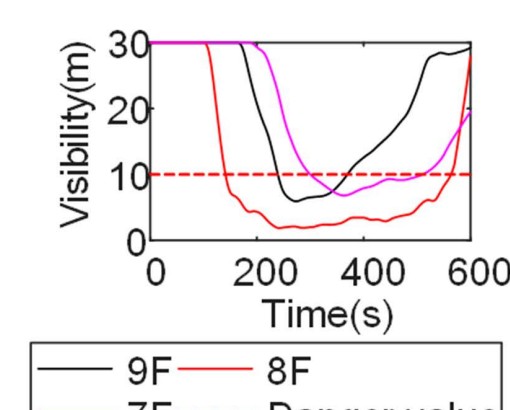

**Fig 23. Visibility in Stairwells in Case 6 over time. (A)** Case6 Stairwell 1. **(B)** Case6 Stairwell 2.

deployed within leeward rescue areas relative to the ignition location, where visibility conditions remain tenable for a longer duration.

## Analysis and discussion

Based on FDS simulations across multiple fire scenarios, fire propagation results enable advanced prediction of fire development trends, which are stored in a dedicated evacuation database, thereby enabling the construction of evacuation spatiotemporal maps. During a fire event, the database provides real-time information on interior and exterior safe areas and dynamically generates evacuation routes for occupants located in different rooms throughout the building. Occupants can access the fire alert system through a mobile application to obtain real-time fire information and input their current locations. The application rapidly generates adaptive evacuation routes, guiding occupants to safety via stairwells or external wall rescue capsules.

A

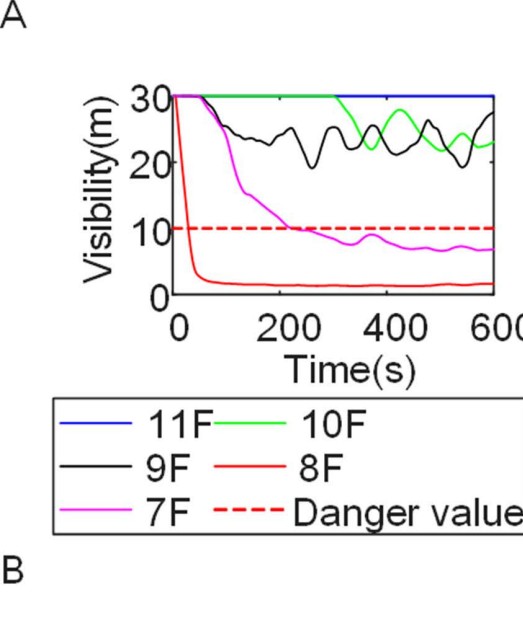

B

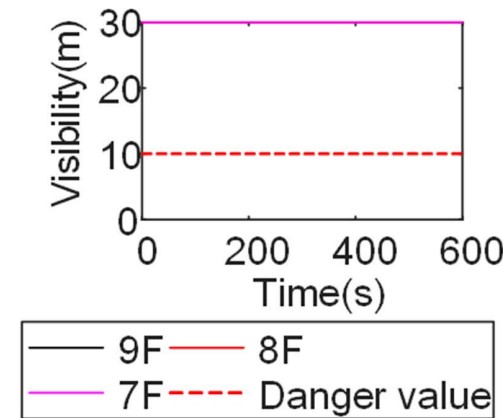

**Fig 24. Visibility in Stairwells in Case 7 over time. (A)** Case7 Stairwell 1. **(B)** Case7 Stairwell 2.

Case 1 represents a typical mid-level fire scenario in which the ignition occurred on the eighth floor in Room 13, adjacent to Stairwell 1, resulting in both stairwells becoming untenable for extended durations. Based on the simulation results for Case 1, a differentiated evacuation strategy is recommended: occupants on the fire floor and lower floors should be evacuated via stairwells, whereas occupants above the fire floor should be evacuated using external rescue capsules. The simulation framework enables identification of the safest evacuation route for each occupant based on their real-time location and evolving fire conditions.

For example, occupants located in Rooms 1, 2, 10, 11, and 12 in Corridor Area A near the initial fire location can evacuate downward via Stairwell 1 within approximately 60 s after fire initiation. Occupants in rooms near Corridor Areas B and C can evacuate within approximately 180 s either by descending via Stairwell 2 or by ascending to higher floors to access rescue capsules. Under dynamic evacuation route guidance, evacuation efficiency is significantly improved, while proactive occupant engagement in the evacuation process is substantially enhanced.

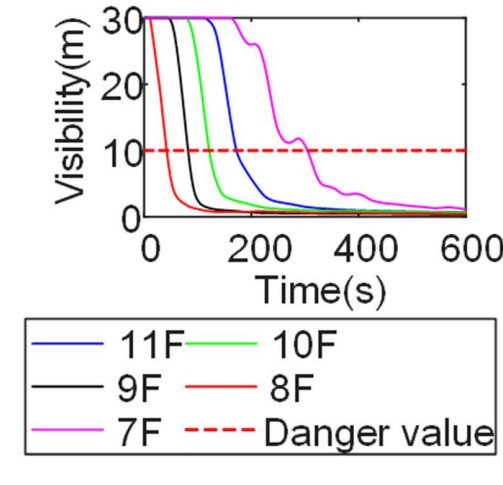

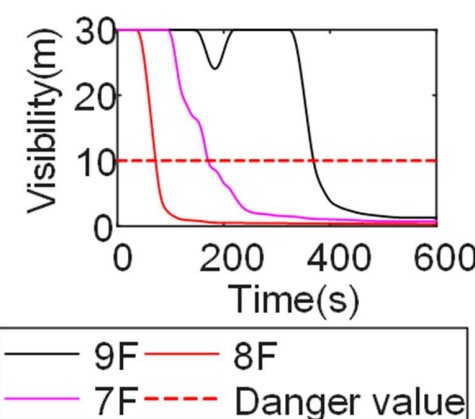

**Fig 25. Visibility in Stairwells in Case 8 over time. (A)** Case8 Stairwell 1. **(B)** Case8 Stairwell 2.

Based on the above analysis, it can be concluded that the L-shaped building geometry induces asymmetric smoke dispersion, and fire spread is the result of the coupled effects of multiple on-site factors. Safe evacuation decisions are directly correlated with the location of trapped occupants, the time of alarm activation, and the rescue method employed. Minor variations in any parameter can lead to fundamental differences in evacuation outcomes. Existing generic, static fire safety evacuation guidance demonstrates significant limitations. Consequently, generating real-time, dynamic escape paths based on actual fireground conditions is essential for improving rescue efficiency and success rates.

## Conclusions

This study established a fire simulation model based on the Fire Dynamics Simulator (FDS) for a 15-story L-shaped building. Numerical simulations were conducted for representative scenarios under worst-case fire conditions, considering various combinations of initial fire location, floor height of ignition, and wind direction. The analysis systematically revealed

A

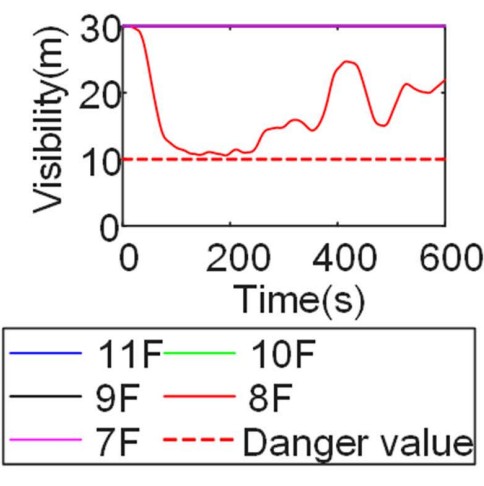

B

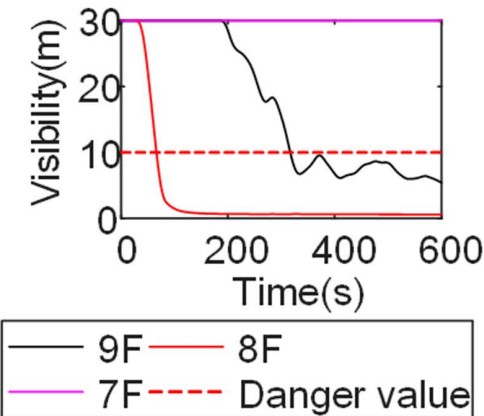

**Fig 26. Visibility in Stairwells in Case 9 over time. (A)** Case9 Stairwell 1. **(B)** Case9 Stairwell 2.

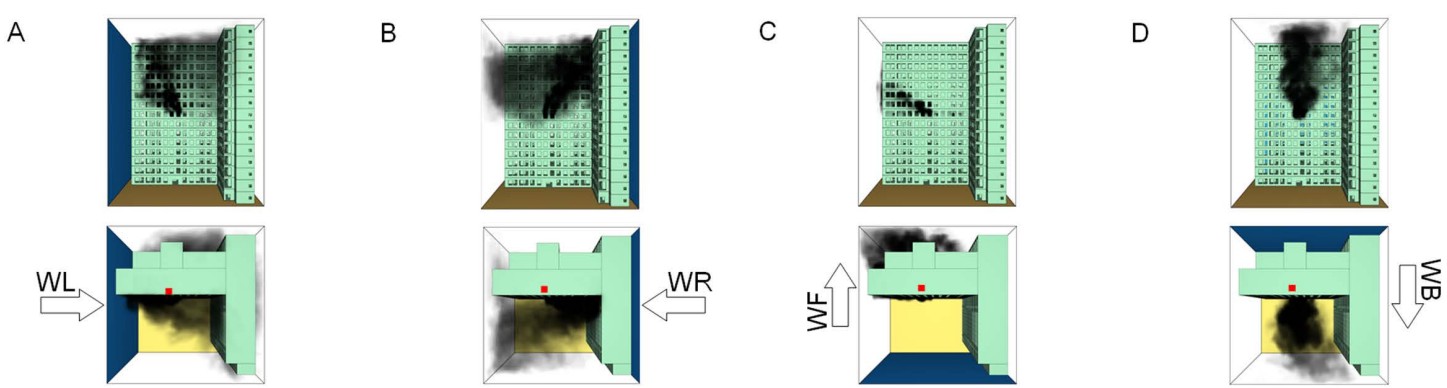

**Fig 27. Live images at 600 seconds (red squares indicate initial fire location). (A)** Case6 Live images. **(B)** Case7 Live images. **(C)** Case8 Live images. **(D)** Case9 Live images.

A

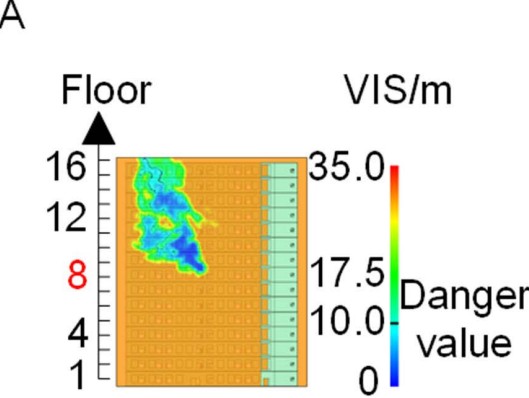

B

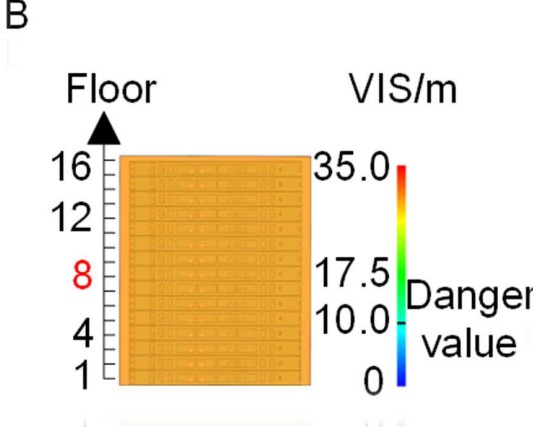

**Fig 28. Visibility of building exteriors in Case 6. (A)** Case6 F-F section. **(B)** Case6 G-G section.

the dynamic spread patterns and evolutionary characteristics of smoke within the fire floor, inside stairwells across different floors, and along the building's external facade. The innovation and core contributions of this research are primarily manifested in the following aspects. First, a multi-factor coupled fire evolution prediction model for L-shaped buildings was developed. This model elucidates the fire development patterns in such structures and clarifies the relationship between fire spread modes and dynamic evacuation decision-making. It not only provides a theoretical basis for generating dynamic escape paths for occupants trapped in different building locations but also lays a scientific foundation for the intelligent dispatch and deployment of emergency rescue systems. Second, "visibility" was explicitly identified as the core criterion for determining occupant safe evacuation in fire scenarios. This addresses limitations in parameter selection inherent in traditional risk assessments and offers a more reliable technical basis for the quantitative evaluation of fire risk and the refinement of emergency decision-making. Third, a generalized, multi-scenario coupled fire evolution simulation framework was proposed. This framework transcends the application constraints associated with a single building type, providing significant scientific support for the formulation, optimization, and engineering implementation of fire safety standards. The findings of this study can be directly applied to fire emergency response operations in actual buildings. A key direction for future research involves integrating the simulation data obtained from the worst-case fire scenarios in

A

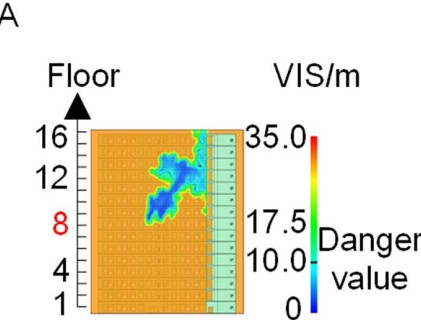

B

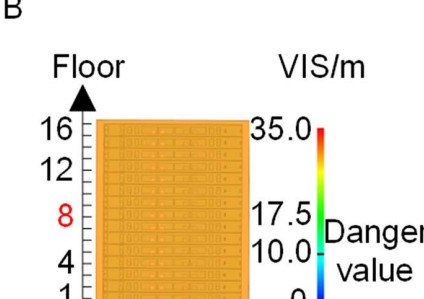

**Fig 29. Visibility of building exteriors in Case 7. (A)** Case7 F-F section. **(B)** Case7 G-G section.

A

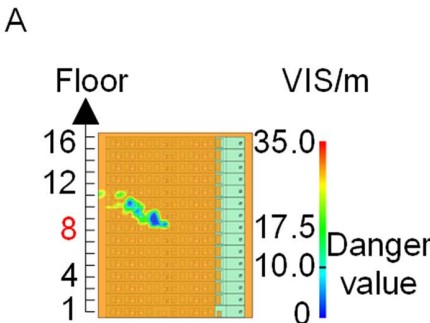

B

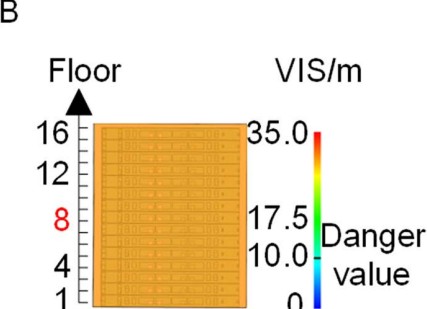

**Fig 30. Visibility of building exteriors in Case 8. (A)** Case8 F-F section. **(B)** Case8 G-G section.

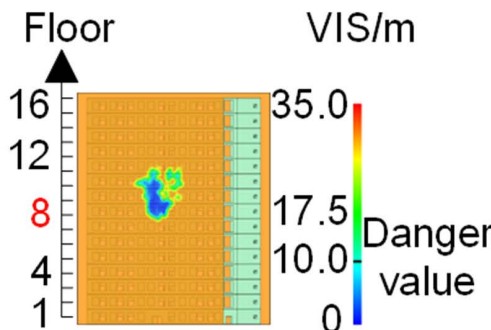

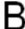

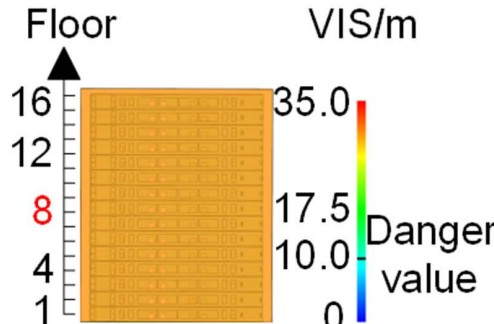

**Fig 31. Visibility of building exteriors in Case 9. (A)** Case9 F-F section. **(B)** Case9 G-G section.

this study with machine learning techniques. This integration aims to construct a spatiotemporal database of building-wide evacuation patterns and to achieve the rapid generation of real-time, dynamic escape routes.

## Supporting information

**S1 File. Case 1–9 Data.**
(RAR)

## Acknowledgments

This section is intended only for general acknowledgements and thanks. Any information related to funding, data availability, author contributions, etc. should be entered directly into their dedicated fields in the PLOS Editorial Manager submission system, which will then be incorporated into the appropriate section in your article during the production process.

## Author contributions

**Conceptualization:** Qin Zhang.

**Data curation:** Yuhong Hu.

**Investigation:** Yuhong Hu.

**Methodology:** Yuhong Hu.

**Project administration:** Qin Zhang, Xiaoju Li.

**Software:** Yuhong Hu.

**Supervision:** Qin Zhang, Xiaoju Li.

**Validation:** Yuhong Hu.

**Visualization:** Yuhong Hu.

**Writing – original draft:** Yuhong Hu.

**Writing – review & editing:** Qin Zhang, Xiaoju Li.

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
