## [Decision Letter · Decision Letter 0]

30 Dec 2025

Dear Dr. Zhang,

Thank you for submitting your manuscript to PLOS ONE. After careful consideration, we feel that it has merit but does not fully meet PLOS ONE’s publication criteria as it currently stands. Therefore, we invite you to submit a revised version of the manuscript that addresses the points raised during the review process.

We look forward to receiving your revised manuscript.

Kind regards,

Gianluca Genovese, Ph.D.

Academic Editor

PLOS One

Journal Requirements:

3. Please update your submission to use the PLOS LaTeX template. The template and more information on our requirements for LaTeX submissions can be found at http://journals.plos.org/plosone/s/latex....

4. Please ensure that you refer to Figures 28, 29, 30, and 31 in your text as, if accepted, production will need this reference to link the reader to the figures.

6. Please upload a copy of all figures, to which you refer in your text. If the figures are no longer to be included as part of the submission please remove all reference to it within the text.

Reviewers' comments:

Reviewer's Responses to Questions

**Comments to the Author**

1. Is the manuscript technically sound, and do the data support the conclusions?

Reviewer #1: Yes

Reviewer #2: Yes

2. Has the statistical analysis been performed appropriately and rigorously?

Reviewer #1: Yes

Reviewer #2: N/A

3. Have the authors made all data underlying the findings in their manuscript fully available?

Reviewer #1: Yes

Reviewer #2: Yes

4. Is the manuscript presented in an intelligible fashion and written in standard English?

Reviewer #1: Yes

Reviewer #2: Yes

Reviewer #1: The manuscript presents a comprehensive numerical investigation of fire scenario evolution and emergency response strategies in an L-shaped high-rise building using FDS. The study addresses a relevant and practical fire safety problem, particularly for complex building geometries that are often underrepresented in existing fire modeling literature. The systematic construction of multiple fire scenarios and the identification of visibility as a key safety-critical parameter are valuable contributions.

However, while the technical execution of the simulations is generally sound, the manuscript would benefit from clearer articulation of its research objectives, stronger justification of modeling assumptions, and a more explicit positioning of its novelty and contribution relative to existing studies. Strengthening these aspects would significantly improve the clarity, scientific rigor, and impact of the work.

Reviewer #2: This paper presents a research on the dynamic evolution of smoke dispersion under varying conditions—including initial fire locations (stairwells or corridor bends), floor height of ignition (low/medium/high zones), and wind direction. However, the description of the abstract needs to be redesigned.

There has been extensive research on building fires. What is the most important innovation point in this paper? What is the basis for setting fire scene parameters?

The article's expression is not standardized and there are some incorrect expressions.

Numerical simulations lack verification of grid independence.

The language description needs improvement.

.

Reviewer #1: No

Reviewer #2: No

---

## [Author Response · Author response to Decision Letter 1]

18 Feb 2026

Dear Editor and Reviewers,

We thank the reviewers for providing constructive feedback. We have fully revised our manuscript (PONE-D-25-63824) and have addressed all of the reviewers’ comments, as well as added new analyses to further strengthen our work. The major revisions and new analyses we have undertaken are summarized in the document titled "Response to Reviewers" and discussed in detail in the point-by-point responses. We have submitted the manuscript file in PDF format and attached the .tex file as “other.”

(Note: Added and amended content in the revised draft has been highlighted in blue, while the primary revisions in the revised draft have been highlighted in yellow.)

---

## [Decision Letter · Decision Letter 1]

25 Mar 2026

Research on Fire Scenario Analysis and Emergency Response Strategies for L-shaped Buildings Using FDS

PONE-D-25-63824R1

Dear Dr. Zhang,

We’re pleased to inform you that your manuscript has been judged scientifically suitable for publication and will be formally accepted for publication once it meets all outstanding technical requirements.

Kind regards,

Gianluca Genovese, Ph.D.

Academic Editor

PLOS One

Reviewers' comments:

Reviewer's Responses to Questions

**Comments to the Author**

Reviewer #2: All comments have been addressed

2. Is the manuscript technically sound, and do the data support the conclusions?

Reviewer #2: Yes

3. Has the statistical analysis been performed appropriately and rigorously?

Reviewer #2: N/A

4. Have the authors made all data underlying the findings in their manuscript fully available?

Reviewer #2: Yes

5. Is the manuscript presented in an intelligible fashion and written in standard English?

Reviewer #2: Yes

Reviewer #2: (No Response)

.

Reviewer #2: No

---

## [Editor Report · Acceptance letter]

PONE-D-25-63824R1

PLOS One

Dear Dr. Zhang,

I'm pleased to inform you that your manuscript has been deemed suitable for publication in PLOS One. Congratulations! Your manuscript is now being handed over to our production team.

Kind regards,

on behalf of

Dr. Gianluca Genovese

Academic Editor

PLOS One